# ZERO-SHOT CONCEPT BOTTLENECK MODELS

## ABSTRACT

Concept bottleneck models (CBMs) are inherently interpretable and intervenable neural network models, which explain their final class label prediction via intermediate predictions of high-level semantic *concepts*. However, they require target task training to learn input-to-concept and concept-to-class mappings, which necessitates collecting target datasets and significant training resources. In this paper, we present *zero-shot concept bottleneck models* (Z-CBMs), which predict concepts and labels in a fully zero-shot manner without additional training of neural networks. Z-CBMs leverage a large-scale concept bank, comprising millions of vocabulary extracted from the web, to describe diverse inputs across various domains. For the input-to-concept mapping, we introduce *concept retrieval*, which dynamically identifies input-related concepts through cross-modal search within the concept bank. In the concept-to-class inference, we apply *concept regression* to select essential concepts from the retrieved concepts by sparse linear regression. Through extensive experiments, we demonstrate that our Z-CBMs provide interpretable and intervenable concepts without any additional training.

## 1 INTRODUCTION

Developing human-interpretable models remains a primary interest within the deep learning research community. Concept bottleneck model (CBM, Koh et al. (2020)) is an inherently interpretable neural network model, which aims to explain its final prediction via the intermediate *concept* predictions. Typically, CBMs are trained end-to-end on a target task to learn the input-to-concept and concept-to-class mappings. A concept consists of high-level semantic vocabulary for describing objects of interest in input data. For instance, CBMs can predict the final label "apple" from the linear combination of the concepts "red sphere," "green leaf," and "glossy surface." These intermediate concept predictions not only provide interpretability but also intervenability in the final prediction by editing the predicted concepts.

In original CBMs (Koh et al., 2020), a concept set for each class label is defined by manual annotations, incurring massive labeling costs greater than ones of the class labels. To reduce these costs, Oikarinen et al. (2023) and Yuksekgonul et al. (2023) automatically generate the concept sets by large language models (LLMs, e.g., GPT-3 (Brown et al., 2020a)) and use the multi-modal embedding space of vision-language models (VLMs, e.g., CLIP (Radford et al., 2021)) to learn the input-to-concept mapping through similarities in the multi-modal feature space. Although these modern CBMs are free from manually pre-defined concepts, their practicality is still restricted by the requirements of training input-to-concept and concept-to-class mappings on target datasets. This means that CBMs have not been available without manually collecting target datasets and additional training of model parameters on them so far. Furthermore, CBMs allow interventions only in static concepts that are used in training, preventing human experts from flexible interactions with arbitrary concepts.

To overcome these limitations, this paper introduces a novel problem setting of CBMs in a zero-shot manner for target tasks. In this setting, we can access pre-trained VLMs, but we cannot know the concepts composing target data in advance. This setting necessitates a two-stage zero-shot inference of input-to-concept and concept-to-class for unseen input samples. The zero-shot input-to-concept inference can not be solved by a naïve application of VLMs as the ordinary zero-shot classification of input-to-label, because it requires identifying a subset of relevant concepts from the large set of all concepts, rather than predicting a single label. Furthermore, the zero-shot concept-to-class inference is difficult because how to obtain the concept-to-class mapping is not obvious without target data and training, which are unavailable in this setting. Therefore, our primary research question is: *How*

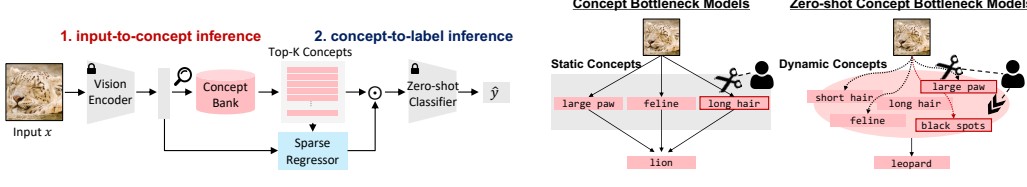

(a) Overview of Z-CBMs    (b) Comparison of Intervention

Figure 1: Zero-shot concept bottleneck models (Z-CBMs). (a) Z-CBMs predict concepts for input by retrieving them from a large-scale concept bank. Then, Z-CBMs predict labels based on the weighted sum of the retrieved concept vectors with importance weights yielded by sparse linear regression. (b) Z-CBMs can be intervened through modifying retrieved concepts and adding arbitrary concepts, whereas conventional CBMs allow only interventions in static concepts. The diverse and dynamic concepts produce accurate predictions and flexible collaborations with human experts.

*can we achieve interpretable and intervenable concept-based prediction through zero-shot input-to-concept and concept-to-class inference without relying on target datasets and additional training?*

We present a novel CBM class called *zero-shot concept bottleneck models* (Z-CBMs). Z-CBMs are zero-shot interpretable models that employ off-the-shelf pre-trained VLMs with frozen weights as the backbone (Fig. 1). Conceptually, Z-CBMs first perform **concept retrieval** to dynamically identify input-related concepts from a broad concept bank and then **concept regression** predicts the final label by simulating zero-shot classification capabilities of black-box VLMs via reconstructing the original input embedding from the retrieved concept embeddings. Our primary contribution is to achieve zero-shot input-to-concept and concept-to-class inference with this framework without additional training. Furthermore, our Z-CBMs allow interventions by arbitrary concepts described in natural language through the VLM feature spaces.

We implement the components of Z-CBMs with simple yet carefully designed and effective techniques. For concept retrieval, Z-CBMs should cover broad domains to provide sufficient concepts for unseen inputs. To cover broad concepts, we build a large-scale concept bank, which is composed of millions of vocabulary extracted from large-scale text caption datasets such as YFCC (Thomee et al., 2016). Given an input sample, Z-CBMs dynamically retrieve concept candidates from the concept bank with an efficient and scalable cross-modal search algorithm. For concept regression, Z-CBMs estimate the importance of concepts for the input feature and then predict labels by the importance-weighted concept features. However, many of the retrieved concept candidates semantically overlap with each other, and thus, the semantically duplicated concepts with high importance by a naïve estimation method can harm the interpretability and intervenability for humans. To overcome this challenge, Z-CBMs find essential and mutually exclusive concepts for the final label prediction by leveraging sparse linear regression (e.g., lasso) to reconstruct the input visual feature vector by a weighted sum of the concept candidate vectors. Combining concept retrieval and concept regression enables Z-CBMs to predict final class labels with interpretable concepts for various domain inputs without any target training.

Our extensive experiments on 12 datasets demonstrate that Z-CBMs can provide interpretable and intervenable concepts without any additional training. Specifically, we demonstrate that the sparse concepts identified by Z-CBMs exhibit strong correlation with input images and cover the annotated concepts in existing training-based CBMs. Furthermore, the Z-CBMs' performance can be enhanced by human intervention in the predicted concepts, emphasizing the reliability of the concept-based prediction. We also show that Z-CBMs competitively perform with black box VLMs and existing CBMs with training. These results suggest the practicality of Z-CBMs for various domains.

## 2 PRELIMINARIES

### 2.1 CONCEPT BOTTLENECK MODELS

A CBM (Koh et al., 2020) is a classifier composed of a concept predictor $g : \mathcal{X} \to \mathcal{C}^K$ and a class label predictor $h : \mathcal{C}^K \to \mathcal{Y}$, where $\mathcal{X}, \mathcal{C}, \mathcal{Y}$ are input, concept, and class label spaces, and $K$ is the number of concepts. The goal is to predict the final class label $y \in \mathcal{Y}$ of input $x \in \mathcal{X}$ based on $K$ interpretable concepts $C = \{c_i \in \mathcal{C}\}_{i=1}^K$. To guarantee the interpretability and classification

performance, $g$ and $h$ are jointly optimized on the following objective function Koh et al. (2020):

$$\min_{g,h} \mathbb{E}_{(x,C,y)\in\mathcal{D}} \left[ \mathcal{L}(h \circ g(x), y) + \alpha \sum_{i}^{K} \mathcal{L}(g(x)_i, c_i) \right], \tag{1}$$

where $\mathcal{D}$ is a training dataset, $\alpha$ is a hyperparameter, and $\mathcal{L}$ is a supervised loss function such as softmax cross-entropy loss. That is, CBMs' interpretability is defined by their ability to detect concepts in the input accurately, which is obtained through the training of input-to-concept and concept-to-class predictions. In this sense, CBMs have the challenge of requiring human annotations of concept labels, which are more difficult to obtain than target class labels. Another challenge is potential performance degradation compared to backbone black-box models (Zarlenga et al., 2022; Moayeri et al., 2023; Xu et al., 2024) due to the difficulty of learning long-tailed concept distributions (Ramaswamy et al., 2023).

## 2.2 Concept Bottleneck Models based on Vision-Language Models

To address the challenges, recent works (Yuksekgonul et al., 2023; Oikarinen et al., 2023; Yang et al., 2023) have focused on leveraging the capabilities of vision-language models (VLMs, e.g., CLIP (Radford et al., 2021)) and large language models (LLMs, e.g., GPT3 (Brown et al., 2020b)). These works automatically generate $C$ in text for each $(x, y) \in \mathcal{D}$ by prompting LLM, and then, train $g$ and $h$ using multi-modal feature spaces with a vision encoder $f_V : \mathcal{X} \to \mathbb{R}^d$ and a text encoder $f_T : \mathcal{T} \to \mathbb{R}^d$ provided by a VLM. We refer to such CBMs based on LLMs and VLMs as VLM-based CBMs. As pioneering works, Post-hoc CBMs (Yuksekgonul et al., 2023), Label-free CBMs (Oikarinen et al., 2023), and LaBo (Yang et al., 2023) firstly implemented this idea. The successor works have assumed the use of LLMs or VLMs, further advancing VLM-based CBMs (Panousis et al., 2023; Rao et al., 2024b; Tan et al., 2024; Srivastava et al., 2024). In particular, Panousis et al. (2023) and Rao et al. (2024a) are related to our work in terms of using sparse modeling to select concepts for input images. However, all of these existing VLM-based CBMs still require training specialized neural networks on target datasets, incurring additional target data collection and training resources.

Handling the bi-level prediction in a zero-shot manner for unseen input is challenging because it can not be solved by naïve application of the existing zero-shot classification methods, which depend on limited vocabularies such as concepts related to ImageNet class names (Norouzi et al., 2014; Demirel et al., 2017; Menon & Vondrick, 2023). Furthermore, current VLM-based CBMs and the recent interpretable framework for CLIP (Bhalla et al., 2024) limit the number of concepts to a few thousand due to training and computational constraints, restricting the generality.

In contrast to the previous VLM-based CBMs, the main purpose of this paper is to achieve fully zero-shot CBMs, which perform inference for input images from various domains without any additional training on target datasets.

# 3 Zero-shot Concept Bottleneck Models

In this section, we formalize the framework of Z-CBMs, which perform a zero-shot inference of input-to-concept and concept-to-class without target datasets and additional training (Fig. 1). Z-CBMs are composed of *concept retrieval* and *concept regression*. Concept retrieval finds a set of the most input-related concept candidates from millions of concepts by querying an input image feature with a semantic similarity search (Fig. 2a). Concept regression estimates the importance scores of the concept candidates by sparse linear regression to reconstruct the input feature (Fig. 2b). Finally, Z-CBMs provide the final label predicted by the reconstructed vector and concept explanations with importance scores.

## 3.1 Zero-shot Inference Algorithm

**Concept Retrieval.** We first find the most semantically related concept candidates to input images from the large spaces in a concept bank $\mathbf{C}$ (Fig. 2a). Given an input $x$, we retrieve the set of $K$ concept candidates $C_x \subset \mathbf{C}$ by using image and text encoders of pre-trained VLMs $f_V$ and $f_T$ as

$$C_x = \underset{c \in \mathbf{C}}{\text{Top-K}} \ \text{Sim}(f_V(x), f_T(c)), \tag{2}$$

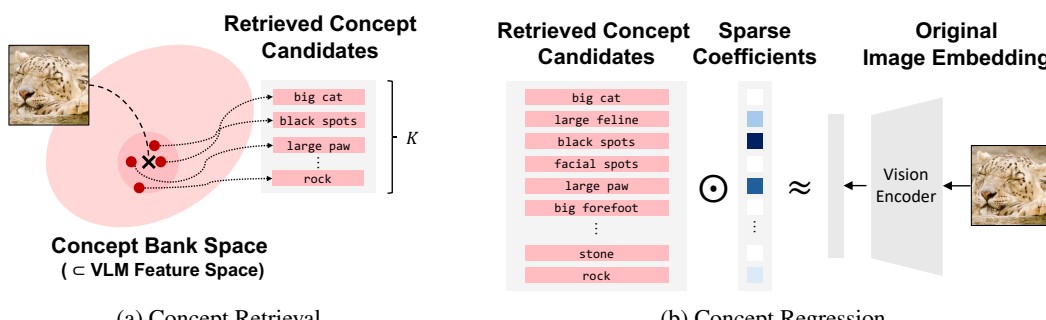

(a) Concept Retrieval  (b) Concept Regression

Figure 2: Concept retrieval and concept regression. (a) Concept retrieval searches concept candidates close to an input image in the VLM feature space and returns the top-$K$ concepts, enabling Z-CBMs to use a large-scale concept bank for general input images. (b) Concept regression selects the important concepts via sparse linear regression, which approximates the input feature vectors by the weighted sum of concept candidate vectors with sparse coefficients. This is helpful in selecting unique concepts.

---

**Algorithm 1** Zero-shot Inference of Z-CBMs

---

**Require:** Input $x$, concept bank $\mathbf{C}$, image encoder $f_{\mathrm{V}}$, text encoder $f_{\mathrm{T}}$
**Ensure:** Predicted label $\hat{y}$, concepts $C_x$, importance weight $W_{C_x}$

1: # Retrieving top-K concepts from input
2: $C_x \leftarrow \underset{c \in \mathbf{C}}{\text{Top-K}}\ \mathrm{Sim}(f_{\mathrm{V}}(x), f_{\mathrm{T}}(c))$
3: $F_{C_x} \leftarrow [f_{\mathrm{T}}(c_1), ..., f_{\mathrm{T}}(c_K)]$
4: # Predicting importance weights by sparse linear regression
5: $W_{C_x} \leftarrow \arg\min_{W \in \mathbb{R}^K} \|f_{\mathrm{V}}(x) - F_{C_x}W\|_2^2 + \lambda\|W\|_1$
6: # Predicting label by importance weighted sum concept vectors
7: $\hat{y} \leftarrow \arg\max_{y \in \mathcal{Y}}\ \mathrm{Sim}(F_{C_x}W_{C_x}, f_{\mathrm{T}}(t_y))$

---

where $\mathrm{Top\text{-}K}$ is an operator yielding top-$K$ concepts in $\mathbf{C}$ from a list sorted in descending order according to a similarity metric $\mathrm{Sim}$. Throughout this paper, we use cosine similarity as $\mathrm{Sim}$ with normalized inputs by following Conti et al. (2023). Thanks to the scalability of the similarity search algorithm (Johnson et al., 2019; Douze et al., 2024), Eq. (2) can efficiently find the concept candidates in an arbitrary concept bank $\mathbf{C}$, which contains millions of concepts to describe inputs in various domains.

**Concept Regression.** Given a concept candidate set $C_x = \{c_1, ..., c_K\}$, we predict the final label $\hat{y}$ by selecting essential concepts from $C_x$. Conventional CBMs infer the $C_x$-to-$\hat{y}$ mapping by training neural regression parameters on target tasks, which incurs the requirements of target dataset collections and additional training. Instead, we solve this task with a different approach, leveraging the zero-shot performance of VLMs. As shown in the previous studies (Radford et al., 2021; Jia et al., 2021), VLMs can be applied to zero-shot classification by inferring a label $\hat{y}$ by matching input $x$ and a class name text $t_y \in \mathcal{T}$ in the multi-modal feature spaces as follows.

$$\hat{y} = \underset{y \in \mathcal{Y}}{\arg\max}\ \mathrm{Sim}(f_{\mathrm{V}}(x), f_{\mathrm{T}}(t_y)). \tag{3}$$

If the feature vector $f_{\mathrm{V}}(x)$ can be approximated by $C_x$, we can achieve the zero-shot performance of black-box features by interpretable concept features. Based on this idea, we approximate $f_{\mathrm{V}}(x)$ by the weighted sum of the concept features $F_{C_x} = [f_{\mathrm{T}}(c_1), ..., f_{\mathrm{T}}(c_K)] \in \mathbb{R}^{d \times K}$ with an importance weight $W \in \mathbb{R}^K$ (Fig. 2b). To obtain $W$, we solve the linear regression problem defined by

$$\min_W \|f_{\mathrm{V}}(x) - F_{C_x}W\|_2^2 + \lambda\|W\|_1. \tag{4}$$

Through this objective, we can achieve $W$ not only for approximating image features but also for effectively estimating the contribution of each concept to the label prediction owing to the sparse regularization $\|W\|_1$. Since $C_x$ is retrieved from a large-scale concept bank $\mathbf{C}$, it often contains noisy concepts that are similar to each other, undermining interpretability due to semantic duplication. In this context, the sparse regularization enhances interpretability by penalizing and eliminating unimportant concepts for the label prediction (Hastie et al., 2015).

Table 1: Concept Accuracy on Bird (Welinder et al., 2010). Z-CBMs can infer large parts of ground-truth concepts without additional training.

| Model | Accuracy |
|---|---|
| CBM (Koh et al., 2020) | 71.61 |
| CDM (Panousis et al., 2023) | 45.61 |
| Z-CBM (Ours) | 60.49 |

**Final Label Prediction.** Finally, we compute the output label with $F_{C_x}$ and $W$ in the same fashion as the zero-shot classification by Eq. (3), i.e.,

$$\hat{y} = \arg\max_{y \in \mathcal{Y}} \text{Sim}(F_{C_x} W, f_{\text{T}}(t_y)). \tag{5}$$

Algorithm 1 shows the overall protocol of the zero-shot inference of Z-CBM. This zero-shot inference algorithm can be applied not only to pre-trained VLMs but also to their linear probing, i.e., fine-tuning a linear head layer on the fixed feature extractor of VLMs for target tasks.

## 3.2 Feasibility Study

We show a preliminary experiment evaluating how much Algorithm 1 can accurately infer the ground-truth concepts, given the concept set from a fully annotated dataset as the concept bank $\mathbf{C}$. To this end, we tested CBM (Koh et al., 2020), CDM (Panousis et al., 2023) as a VLM-based CBM, and Z-CBM on the Bird dataset (Welinder et al., 2010), which has human-annotated concept labels. We used CLIP ViT-B/32 (Radford et al., 2021) as the backbone and the annotated 312 concepts as the concept bank $\mathbf{C}$ for CDM and Z-CBM. More detailed protocols are described in Appendix A. Table 1 shows the concept accuracy. Z-CBM outperformed CDM, which requires additional training, and achieved approximately 80% of the performance of CBM trained with the ground truth concepts. This indicates that Z-CBMs can find valid concepts without additional training by concept retrieval and regression. In practice, since full concept annotations are not generally given for unseen inputs, Z-CBMs adopt a large-scale concept bank that covers broad concepts for various domains.

## 4 Implementation

In this section, we present the detailed implementations of Z-CBMs, including backbone VLMs, concept bank construction, concept retrieval, and concept regression.

**Vision-Language Models.** Z-CBMs allow to leverage arbitrary pre-trained VLMs for $f_{\text{V}}$ and $f_{\text{T}}$. We basically use the official implementation of OpenAI CLIP (Radford et al., 2021) and the publicly available pre-trained weights.[1] Specifically, by default, we use ViT-B/32 as $f_{\text{V}}$ and the base transformer with 63M parameters as $f_{\text{T}}$ by following the original CLIP. In Section 5.6.1, we show that other VLM backbones (e.g., SigLIP (Zhai et al., 2023) and OpenCLIP (Cherti et al., 2023)) are also available for Z-CBMs.

**Concept Bank Construction.** Here, we introduce the construction protocols of the concept bank $\mathbf{C}$ of Z-CBMs. As Z-CBMs operate without prior knowledge of input concepts, the concept bank must possess a sufficient vocabulary to describe the inputs from diverse domains. To this end, we extract concepts from multiple image caption datasets and integrate them into a single concept bank. Specifically, we automatically collect concepts as noun phrases by parsing each sentence in the caption datasets including Flickr-30K (Young et al., 2014), CC-3M (Sharma et al., 2018), CC-12M (Changpinyo et al., 2021), and YFCC-15M (Thomee et al., 2016); we use the parser implemented in `nltk` (Bird, 2006). At this time, the concept set size is $|C| \approx 20\text{M}$. Then, we filter the large base concept set to remove nonessential concepts, following policies of Oikarinen et al. (2023); please see Appendix B. Finally, after filtering concepts, we obtain the concept bank containing $|C| \approx 5\text{M}$ concepts. We also discuss the effect of varying caption datasets used for collecting concepts in Sec. 5.5 and 5.6.2.

**Similarity Search in Concept Retrieval.** Concept retrieval searches the concept candidates from input feature vectors. To this end, we implement the concept search component by the open source

---

[1]https://github.com/openai/CLIP

Table 2: SigLIP-Score

| Method | Avg. of 12 datasets |
|---|---|
| Label-free CBM | 0.5485 |
| LaBo | 0.5419 |
| CDM | 0.5714 |
| Z-CBM (ALL) | **0.6309** |

Table 3: Concept Recall (%)

| Method | Avg. of 12 datasets |
|---|---|
| Z-CBM (Cosine Similarity) | 58.51 |
| Z-CBM (Linear Regression) | 76.87 |
| Z-CBM (Lasso) | **85.27** |

library of Faiss (Johnson et al., 2019; Douze et al., 2024). First, we create a search index based on the text feature vectors of all concepts in a concept bank $\mathbf{C}$ using $f_\mathrm{T}$. At inference time, we retrieve the concept vectors via similarity search on the concept index by specifying the concept number $K$. We set $K = 2048$ as the default value and empirically show the effect of $K$ in Appendix D.5.

**Sparse Linear Regression in Concept Regression.** In concept regression, we can use arbitrary sparse linear regression algorithms, including lasso (Tibshirani, 1996), elastic net (Zou & Hastie, 2005), and sparsity-constrained optimization like hard thresholding pursuit (Yuan et al., 2014). The efficient implementations of these algorithms are publicly available on the `sklearn` (Pedregosa et al., 2011) and `skscope` (Wang et al., 2024) libraries. The choice of sparse linear regression algorithm depends on the use cases. For example, lasso is beneficial for naturally extracting important concepts from a large number of candidate concepts; elastic net is effective for maximizing target performance; and sparsity-constrained optimization allows for strict control over the number of concepts used in explanations. We use lasso with $\lambda = 1.0 \times 10^{-5}$ as the default algorithm (see Appendix C and D.3), but we confirm that arbitrary sparse linear regression algorithms are available for Z-CBMs in Appendix D.4.

## 5 EXPERIMENTS

We evaluate Z-CBMs on multiple visual classification datasets and pre-trained VLMs. We test two scenarios: *zero-shot*, where pre-trained VLMs perform inference without training, and *training head*, where only classification heads are trained.

### 5.1 SETTINGS

**Datasets.** We evaluated Z-CBMs on 12 diverse image classification datasets: **Aircraft (Air)** (Maji et al., 2013), **Bird** (Welinder et al., 2010), **Caltech-101 (Cal)** (Fei-Fei et al., 2004) **Car** (Krause et al., 2013), **DTD** (Cimpoi et al., 2014), **EuroSAT (Euro)** (Helber et al., 2019), **Flower (Flo)** (Nilsback & Zisserman, 2008), **Food** (Bossard et al., 2014), **ImageNet (IN)** (Russakovsky et al., 2015), **Pet** (Parkhi et al., 2012), **SUN397** (Xiao et al., 2010), and **UCF-101** (Soomro, 2012). They are often used to evaluate the zero-shot generalization performance of VLMs (Radford et al., 2021; Zhou et al., 2022). In the training head scenario, we randomly split a training dataset into $9 : 1$ and used the former as the training set and the latter as the validation set. For ImageNet, we set the split ratio $99 : 1$.

**Zero-shot Baselines.** For the zero-shot baseline, our Z-CBMs with the zero-shot inference of a black-box VLM and ConSe (Norouzi et al., 2014), which is a zero-shot classification method predicting a class label with a weighted sum of ImageNet concept features (see Appendix C).

**Training Head Baselines.** To compare Z-CBMs with existing VLM-based CBMs, we evaluated models trained on target datasets. In this setting, Z-CBMs were applied to linear probing of VLMs, i.e., fine-tuning only a linear head layer on the feature extractors of VLMs; we refer to this pattern LP-Z-CBM. As the baselines, we used **Lable-free CBM** (Oikarinen et al., 2023), **LaBo** (Yang et al., 2023), and **CDM** (Panousis et al., 2023). We performed these methods based on their repositories.

**Evaluation Metrics.** To evaluate predicted concepts, we used the **SigLIP-Score**, the cosine similarity between image and text embeddings on SigLIP (Zhai et al., 2023) (higher is better). This score indicates how well a predicted concept explains an image (Radford et al., 2021; Hessel et al., 2021), serving as a quality indicator for input-to-concept inference. Specifically, we averaged SigLIP-Scores between test images and their predicted concept texts for the top 10 concepts, ranked by absolute importance scores. We also used **concept recall** to evaluate Z-CBM's predicted concepts. Top-$K$ concept recall $|C^\mathrm{Z} \cap C^\mathrm{R}|/K$ measures the overlap between Z-CBM's top-$K$ concepts $C^\mathrm{Z} = \{c_i^\mathrm{Z}\}_{i=1}^{K} \subset \mathbf{C}$ (with non-zero coefficients) and $N_\mathrm{R}$ reference concepts $C^\mathrm{R} = \{c_i^\mathrm{R}\}_{i=1}^{N_\mathrm{R}} \subset \mathbf{C}$

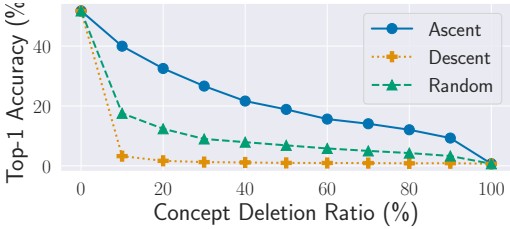

Figure 3: Concept Deletion

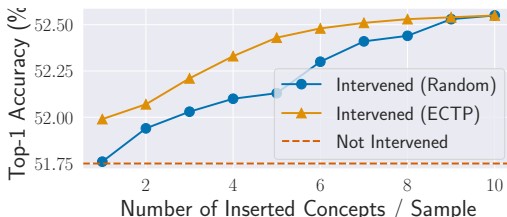

Figure 4: Concept Insertion

from VLM-based CBMs requiring training. This metric assesses concept overlap between Z-CBMs and a training-based CBM using the same concept bank $\mathbf{C}$, indicating Z-CBM's approximation of a trained model's concepts. Specifically, we averaged concept recall at $K = 10$ across test samples, using the GPT-generated concept banks (Oikarinen et al., 2023), and reference concepts of Label-free CBMs Following Oikarinen et al. (2023), $C^{\mathrm{R}}$ comprised concepts with contribution scores $> 0.05$. Finally, we report **top-1 test accuracy** for target classification performance.

## 5.2 Quantitative Evaluation of Predicted Concepts

First, we quantitatively evaluate Z-CBM's predicted concepts for their factual representation of image features. We measure average SigLIP scores and concept recall across 12 datasets.

Table 2 shows the SigLIP-Score results. Across all datasets, our Z-CBM predicted concepts strongly correlated with input images and largely outperformed the training-required VLM-based CBMs. This can be caused by the concept bank choice. Existing VLM-based CBMs perform concept-to-class inference with learnable parameters, making it difficult to handle millions of concepts simultaneously, and thus, limiting their vocabularies to a few thousand for learnability. In contrast, our Z-CBMs can manage millions of concepts without training by dynamically retrieving relevant concepts and inferring essential ones with sparse linear regression. Paradoxically, Z-CBMs achieve accurate image explanations via an abundant concept vocabulary by eliminating training.

Table 3 shows concept recall results using concepts predicted by Label-free CBMs as reference. It also lists results for Z-CBMs using cosine similarity on CLIP and linear regression, instead of lasso, to compute importance coefficients. Z-CBMs with lasso achieved the best concept recall (85.27%). This demonstrates that Z-CBMs can predict most of the important concepts found by VLM-based CBMs, and that sparse linear regression is a key factor for identifying these concepts without training.

## 5.3 Evaluation of Human Intervention

Human intervention in concepts is an essential feature shared by the CBM family for debugging models and modifying the output to make the final prediction accurate. In addition to interventions in existing concepts in the concept bank, Z-CBMs allows interventions in arbitrary concepts described in natural language. We evaluate Z-CBMs via two types of intervention: (i) concept deletion and (ii) concept insertion. In concept deletion, we confirm the dependence on the predicted concepts by removing the concept with non-zero coefficients in ascending, descending, and random orders. Fig. 3 shows the results on Bird by varying the deletion ratio. The accuracy of Z-CBMs largely dropped with the smaller deletion ratio in the descent cases, indicating that Z-CBM selects the important concepts via concept regression and relies on them for the final prediction. In the ascent cases, the accuracy slowly and steadily decreases, suggesting that the Z-CBMs are not biased toward limited concepts and that all of the selected concepts are essential.

In concept insertion, we first predict concepts by concept regression and add randomly selected ground-truth concepts to the output non-zero concept set. Then, we re-run concept regression with linear regression on this modified concept set and predict the final label prediction by Eq. (5). As the ground truth concepts, we used the attribute labels of Bird (Welinder et al., 2010). Fig. 4 shows the top-1 accuracy of the intervened Z-CBMs. In addition to random selection, we performed a sophisticated intervention method called ECTP (Shin et al., 2023). The performance improved as the number of inserted concepts per sample increased for both cases. This indicates that Z-CBMs can correct the final output by modifying the concept of interest through intervention.

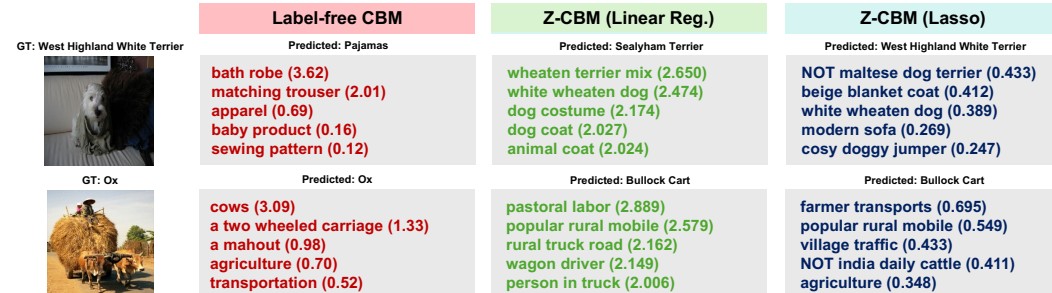

Figure 5: Qualitative evaluation of predicted concepts on the ImageNet validation set. While Label-free CBMs sometimes hallucinate invisible concepts or ignore important concepts, Z-CBMs with lasso provide realistic and dominant concepts in inputs with diverse vocabulary. **NOT** prefix denotes that the concept has negative coefficients similar to Oikarinen et al. (2023) and Panousis et al. (2023).

### 5.4 QUALITATIVE EVALUATION OF PREDICTED CONCEPTS

We demonstrate the qualitative evaluation of predicted concepts by Label-free CBMs and Z-CBMs when inputting the ImageNet validation examples in Fig. 5; we also show the results of Z-CBMs using linear regression to compute the importance coefficients instead of lasso. Overall, Z-CBMs tend to accurately predict realistic and dominant concepts that appear in input images, even though they are not trained on target tasks. For instance, in the first row, Z-CBM predicts various concepts related to dogs, clothes, and background, whereas Label-free CBM focuses on clothes and ignores dogs and background. This difference may be caused by the fact that the image-to-concept mapping of Z-CBMs is not biased toward the label information because it does not train on the target data. Conversely, like the second row, Z-CBMs tend to concentrate on global regions and miss the concepts in local regions; this can be alleviated by intervening in the concept prediction (see Sec. 5.3).

For the comparison of linear regression and lasso, Z-CBM (Linear Reg.) tends to produce semantically overlapped concepts. In fact, quantitatively, we also found that the averaged inner SigLIP-Scores among the top-10 concepts of lasso (0.5552) is significantly lower than that of linear regression (0.7425). These results emphasize the advantage of using sparse modeling in concept regression to select mutually exclusive concepts from the large concept bank.

### 5.5 ZERO-SHOT IMAGE CLASSIFICATION PERFORMANCE

Table 4 shows averaged top-1 accuracy across the 12 datasets, including concept bank ablations (indicated by brackets in Z-CBMs rows). In the zero-shot setting, Z-CBMs unexpectedly outperformed the zero-shot CLIP baseline. This may be because Z-CBMs approximate image features with the weighted sum of textual concept features, reducing the modality gap between the image and the class label text (see Appendix D.2). For the concept bank ablation, larger concept banks yielded higher accuracy, suggesting better image feature approximation with richer vocabularies (further explored Sec. 5.6.2).

In training head, LP-Z-CBMs matched linear probing accuracy and consistently outperformed existing VLM-based CBMs requiring additional training. This implies that Z-CBM's concept retrieval and regression with original CLIP features suffice for effective input-to-concept and concept-to-class inference regarding target task performance.

### 5.6 DETAILED ANALYSIS

#### 5.6.1 EFFECTS OF BACKBONE VLMS

We show the impacts on Z-CBMs when varying backbone VLMs using OpenCLIP (Cherti et al., 2023) and DFN (Fang et al., 2024). Table 5 demonstrates the Z-CBMs' compatibility with diverse backbone VLMs. Z-CBM performance scaled with VLM zero-shot performance. Notably, improved SigLIP-Scores with stronger VLMs indicate more accurate input-to-concept inference. The previously noted outperformance of black-box baselines (Sec. 5.5) was more pronounced in smaller models with weaker multi-modal alignment. These results suggest that Z-CBM is universally applicable across VLM generations and that its practicality will improve as VLMs evolve in future work.

Table 4: Top-1 accuracy on 12 classification datasets with CLIP ViT-B/32. Complete results appear in Appendix.

| Setting | Method | Avg. of 12 datasets |
|---|---|---|
| Zero-Shot | Zero-shot CLIP | 53.73 |
| | ConSe | 10.82 |
| | Z-CBM (Flickr30K) | 52.62 |
| | Z-CBM (CC3M) | 52.98 |
| | Z-CBM (CC12M) | 53.97 |
| | Z-CBM (YFCC15M) | 53.94 |
| | Z-CBM (ALL) | **54.28** |
| Training Head | Linear Probe CLIP | **78.98** |
| | Label-free CBM | 74.87 |
| | LaBo | 74.04 |
| | CDM | 76.39 |
| | LP-Z-CBM (ALL) | 78.31 |

Table 5: Performance of Z-CBMs varying backbone VLMs on ImageNet.

| Backbone VLM | Top-1 Acc. (Black Box) | Top-1 Acc. (Z-CBM) | SigLIP-Score (Z-CBM) |
|---|---|---|---|
| CLIP ViT-B/32 | 61.88 | 62.70 | 0.6498 |
| CLIP ViT-L/14 | 72.87 | 73.19 | 0.6608 |
| OpenCLIP ViT-H/14 | 77.20 | 77.81 | 0.6790 |
| OpenCLIP ViT-G/14 | 79.03 | 78.27 | 0.6810 |
| DFN ViT-H/14 | 83.85 | 83.40 | 0.7038 |

Table 6: Performance of Z-CBMs varying concept banks on ImageNet with CLIP ViT-B/32.

| Concept Bank | Vocab. Size | Top-1 Acc. | SigLIP-Score |
|---|---|---|---|
| Zero-shot CLIP | N/A | 61.88 | N/A |
| Label-free CBM w/ GPT-3 (ImageNet Class) | 4K | 58.00 | 0.5896 |
| CDM w/ GPT-3 (ImageNet Class) | 4K | 62.52 | 0.6193 |
| GPT-3 (ImageNet Class) | 4K | 59.18 | 0.5407 |
| Noun Phrase (Flickr30K) | 45K | 61.52 | 0.5539 |
| Noun Phrase (CC3M) | 186K | 62.38 | 0.5904 |
| Noun Phrase (CC12M) | 2.58M | 62.42 | 0.6242 |
| Noun Phrase (YFCC15M) | 2.20M | 62.45 | 0.6375 |
| Noun Phrase (ALL) | **5.12M** | **62.70** | **0.6498** |

### 5.6.2 EFFECTS OF CONCEPT BANK

As shown in Sec. 5.5 and Table 4, the choice of concept bank is crucial for the performance. Here, we provide a more detailed analysis of the concept banks. Table 6 summarizes the results when varying concept banks. For comparison, we added the concept bank generated by GPT-3 from ImageNet class names in Label-free CBMs (Oikarinen et al., 2023). Z-CBMs with the GPT-3 concepts significantly degraded the top-1 accuracy from Zero-shot CLIP, and the SigLIP-Score was much lower than that of our concept banks composed of noun phrases extracted from caption datasets. This indicates that the concept bank used in the existing method is limited in its ability to represent image concepts. Meanwhile, our concept bank scalably improved in accuracy and SigLIP-Score as its size increased, and combining all of them achieved the best results. We also examine the relationship between the concept bank and target datasets in Appendix D.8.

## 6 CONCLUSION

This paper introduced zero-shot concept bottleneck models (Z-CBMs), a novel framework for predicting input-to-concept and concept-to-class mappings in a fully zero-shot manner. Z-CBMs first search input-related concept candidates by concept retrieval, which leverages pre-trained VLMs and a large-scale concept bank containing millions of concepts to explain outputs for unseen input images in various domains. For the concept-to-class inference, concept regression estimates the importance of concepts by solving the sparse linear regression, approximating the input image features with linear combinations of selected concepts. Our extensive experiments show that Z-CBMs can provide interpretable and intervenable concepts comparable to conventional CBMs that require training. Since Z-CBMs can be built on any off-the-shelf VLMs, it will be a good baseline for zero-shot interpretable models based on VLMs in future research. One limitation is the reliance on off-the-shelf VLMs and pre-defined concept banks for explanations, which may struggle in domain-specific applications such as medical imaging. While we can overcome this limitation by introducing domain-specific VLMs and vocabulary (Wang et al., 2022), developing an explainable model with the versatility to handle any application without providing such prior knowledge is an open question.

## REPRODUCIBILITY STATEMENT

We describe the implementation details in Section 4 and Appendix A, B, and C. We also provide code to reproduce experiments in the supplementary materials.

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

Table 7: SigLIP-Score on 12 classification datasets. We compute the averaged SigLIP-Scores between images and concepts with top-10 absolute coefficients.

| Method | Air | Bird | Cal | Car | DTD | Euro | Flo | Food | IN | Pet | SUN | UCF | Avg. |
|---|---|---|---|---|---|---|---|---|---|---|---|---|---|
| Label-free CBM | 0.6824 | 0.7818 | 0.7023 | 0.7106 | 0.6552 | 0.6179 | 0.6988 | 0.6959 | 0.7202 | 0.7119 | 0.7327 | 0.6688 | 0.6982 |
| LaBo | 0.6980 | 0.7626 | 0.7211 | 0.7411 | 0.6299 | 0.6202 | 0.7138 | 0.7526 | 0.7272 | 0.7235 | 0.7060 | 0.6978 | 0.7078 |
| CDM | 0.6887 | 0.7655 | 0.7164 | 0.7221 | 0.7000 | 0.6584 | 0.7239 | 0.7151 | 0.7618 | 0.7257 | 0.7049 | 0.6870 | 0.7141 |
| Z-CBM (ALL) | **0.7811** | **0.8100** | **0.7748** | **0.7582** | **0.7661** | **0.7457** | **0.7767** | **0.7785** | **0.7766** | **0.7477** | **0.7925** | **0.7965** | **0.7754** |

# A    DETAILS OF FEASIBILITY STUDY IN SECTION 3.2

For all models, we used CLIP ViT-B/32 as the backbone. We implemented CBMs on top of the CLIP visual feature extractor as two linear layer classifiers for input-to-concept and concept-to-class. We trained the CBM by Eq. (1) for 100 epochs while freezing the CLIP feature extractor. We implemented and trained the CDM by following the official implementation by Panousis et al. (2023). Note that we used the concept bank $\mathbf{C}$ containing human-annotated ground-truth concepts of the training set of Bird (Welinder et al., 2010) for all models. We evaluated concept accuracy on Bird using the ground-truth concept labels on the test set. We measured the average precision of ground-truth concepts, which were included among the top 10 important concepts.

# B    DETAILS OF CONCEPT FILTERING

We basically follow the policies introduced by Oikarinen et al. (2023), which removes (i) too long concepts, (ii) too similar concepts to each other, and (iii) too similar concepts to target class names (optional). However, the second policy is computationally intractable because it requires the $\mathcal{O}(|C|^2)$ computation of the similarity matrix across all concepts. Thus, we approximate this using a similarity search by Eq. (2) that yields the most similar concepts. We retrieve the top 64 concepts from a concept and remove them according to the original policy.

# C    DETAILS OF SETTINGS

**Zero-shot Baselines.**    For the black-box baseline, according to the previous work (Radford et al., 2021), we construct a class name prompt $t_y$ by the scheme of "`a photo of [class name]`", and make VLMs predict a target label $\hat{y}$ by Eq. (3). ConSe is a zero-shot cross-modal classification method that infers a target label from a semantic embedding composed of the weighted sum of concepts of the single predicted ImageNet label. For Z-CBMs, we selected $1.0 \times 10^{-5}$ as $\lambda$ by searching from $\{1.0 \times 10^{-2}, 1.0 \times 10^{-3}, 1.0 \times 10^{-4}, 1.0 \times 10^{-5}, 1.0 \times 10^{-6}, 1.0 \times 10^{-7}, 1.0 \times 10^{-8}\}$ to choose the minimum value achieving over 10% non-zero concept ration when using $K = 2048$ on the subset of ImageNet training set. We used the same $\lambda$ for all experiments.

**Reproducibility Statement.**    As described in Sec. 4 and 5 , the implementation of the proposed method uses a publicly available code base. For example, the VLMs backbones are publicly available in the OpenAI CLIP[2] and Open CLIP[3] GitHub repositories. All datasets are also available on the web; see the references in Sec. 5.1 for details. For the computation resources, we used a 24-core Intel Xeon CPU with an NVIDIA A100 GPU with 80GB VRAM. More details of our implementation can be found in the attached code in the supplementary materials and we will make the code available on the public repository if the paper is accepted.

# D    ADDITIONAL EXPERIMENTS

## D.1    DETAILED RESULTS FOR ALL DATASETS

Table 7, 8, and 9 shows all of the results on the 12 datasets omitted in Table 2, 3, and 4, respectively.

---

[2]https://github.com/openai/CLIP

[3]https://github.com/mlfoundations/open_clip

Table 8: Concept Recall (%) of Z-CBMs on 12 classification datasets

| Method | Air | Bird | Cal | Car | DTD | Euro | Flo | Food | IN | Pet | SUN | UCF | Avg. |
|---|---|---|---|---|---|---|---|---|---|---|---|---|---|
| Z-CBM (Cosine Similarity) | 66.83 | 41.42 | 37.13 | 60.95 | 71.85 | 90.37 | 50.39 | 77.50 | 48.80 | 90.07 | 29.76 | 37.04 | 58.51 |
| Z-CBM (Linear Regression) | 96.45 | 81.98 | 51.82 | 58.06 | 91.40 | 90.91 | 90.82 | 90.88 | 71.51 | 95.37 | 40.84 | 62.43 | 76.87 |
| Z-CBM (Lasso) | **98.95** | **86.01** | **69.97** | **96.43** | **94.26** | **91.91** | **93.57** | **96.74** | **86.92** | **97.37** | **42.86** | **68.20** | **85.27** |

Table 9: Top-1 accuracy on 12 classification datasets with CLIP ViT-B/32.

| Setting | Method | Air | Bird | Cal | Car | DTD | Euro | Flo | Food | IN | Pet | SUN | UCF | Avg. |
|---|---|---|---|---|---|---|---|---|---|---|---|---|---|---|
| | Zero-shot CLIP | 18.93 | 51.80 | 24.50 | 60.38 | 43.24 | 35.54 | 63.41 | 78.61 | 61.88 | 85.77 | 61.21 | 59.48 | 53.73 |
| Zero-Shot | Z-CBM (Flickr30K) | 18.27 | 46.70 | 24.26 | 56.46 | 43.56 | 34.32 | 59.80 | 78.17 | 61.52 | 85.46 | 62.23 | 60.67 | 52.62 |
| | Z-CBM (CC3M) | 18.09 | 48.53 | 24.30 | 55.58 | 43.51 | 35.09 | 61.44 | 78.89 | 62.68 | 85.29 | 62.18 | 60.45 | 52.98 |
| | Z-CBM (CC12M) | 18.66 | 51.03 | 24.42 | **59.22** | 43.72 | **36.73** | 63.31 | 79.26 | 62.42 | **85.98** | 62.11 | 60.75 | 52.98 |
| | Z-CBM (YFCC15M) | 18.81 | **51.87** | 24.54 | 58.72 | 43.40 | 35.96 | 63.38 | 79.22 | 62.42 | 85.94 | 62.07 | 60.96 | 53.97 |
| | Z-CBM (ALL) | **19.00** | 51.75 | **25.42** | 58.87 | **43.86** | 36.12 | **63.78** | 82.44 | **62.70** | 85.95 | **62.89** | 61.49 | **54.28** |
| | Linear Probe CLIP | **45.06** | **72.72** | 95.70 | 79.75 | 74.84 | 92.99 | 94.02 | 87.06 | 68.54 | 88.72 | 65.20 | 83.14 | 78.98 |
| Training Head | Label-free CBM | 42.72 | 67.05 | 94.12 | 71.81 | **74.31** | 91.30 | 91.23 | 81.91 | 58.00 | 83.29 | 62.00 | 80.68 | 74.87 |
| | LaBo | 43.43 | 69.38 | 94.82 | 77.78 | 73.59 | 88.17 | 91.67 | 84.29 | 59.16 | 87.24 | 57.70 | 81.26 | 74.04 |
| | CDM | 44.58 | 69.75 | 95.78 | 77.27 | 74.80 | **92.16** | 92.99 | 81.85 | 62.52 | 86.59 | 56.48 | 81.93 | 76.39 |
| | LP-Z-CBM (ALL) | 44.80 | 71.67 | 95.50 | 78.09 | 73.94 | 91.22 | 93.28 | 86.73 | 67.99 | 88.58 | 65.53 | 82.37 | 78.31 |

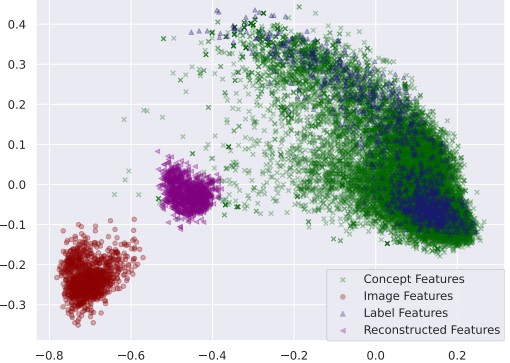

Figure 6: PCA feature visualization of Z-CBMs

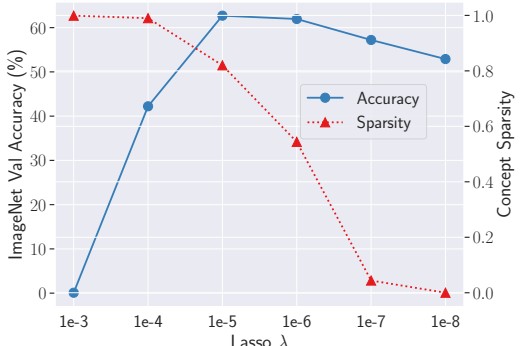

Figure 7: Effects of varying $\lambda$ in Eq. 4

## D.2 ANALYSIS ON MODALITY GAP

In Section 5.5, Table 4 shows that Z-CBMs improved the zero-shot CLIP baselines. We hypothesize that the reason is reducing the modality gap (Liang et al., 2022) between image and text features by the weighted sum of concept features to approximate $f_V(x)$ by Eq. 4. To confirm this, we conduct a deeper analysis of the effects of Z-CBMs on the modality gap with quantitative and qualitative evaluations. For quantitative evaluation, we measured the L2 distance between image-label features and concept-label features as the modality gap by following (Liang et al., 2022). The L2 distances were $1.74 \times 10^{-3}$ in image-to-label and $0.86 \times 10^{-3}$ in concept-to-class, demonstrating that Z-CBMs largely reduce the modality gap by concept regression. We also show the PCA feature visualizations in Figure 6, indicating that the weighted sums of concepts (reconstructed concepts) bridge the image and text modalities.

## D.3 EFFECTS OF $\lambda$

Here, we discuss the effects when changing $\lambda$ in Eq. (4). We varied $\lambda$ in $\{1.0 \times 10^{-2}, 1.0 \times 10^{-3}, 1.0 \times 10^{-4}, 1.0 \times 10^{-5}, 1.0 \times 10^{-6}, 1.0 \times 10^{-7}, 1.0 \times 10^{-8}\}$. Figure 7 plots the accuracy and the sparsity of predicted concepts on ImageNet. Using different lambda varies the sparsity and accuracy. Therefore, selecting appropriate $\lambda$ is important for achieving both high sparsity and high accuracy.

## D.4 EFFECTS OF CONCEPT REGRESSOR

Z-CBMs accommodate various sparse linear regression algorithms, as discussed in Sec. 4. Here, we compare the performance of Z-CBMs with multiple sparse linear regression algorithms: lasso (Tibshirani, 1996), elastic net (Zou & Hastie, 2005), and sparsity-constrained optimization with HTP (Yuan et al., 2014). Further, we evaluate these sparse algorithms by comparing them with non-sparse

Table 10: Performance of Z-CBMs varying concept regressor on ImageNet with CLIP ViT-B/32.

| Concept Regressor | Top-1 Acc. | Sparsity | SigLIP-Score |
|---|---|---|---|
| CLIP Similarity | 14.66 | 0.0000 | 0.5106 |
| Linear Regression | 52.88 | 0.0000 | 0.5563 |
| Lasso | 62.70 | 0.8201 | 0.6498 |
| Elastic Net | 62.84 | 0.7311 | 0.6511 |
| Sparsity-Constrained (HTP) | 62.54 | 0.8750 | 0.6245 |

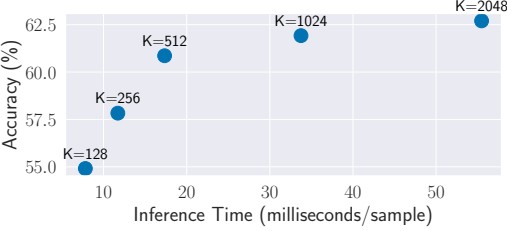

Figure 8: Accuracy vs. inference time by varying retrieved concept number $K$.

algorithms to compute the importance of concepts: CLIP Similarity, which uses the cosine similarity computed on CLIP as the importance, and linear regression. Table 10 shows the performance, where sparsity is a ratio of zero importance coefficients to the total number of concept candidates. While the sparse linear regression algorithms achieved top-1 accuracy scores at the same level, the non-sparse algorithms failed to accurately predict labels from importance-weighted concepts. Additionally, linear regression has unstable numerical computation due to the rank deficiency of the Gram matrix of $F_{C_x}$ when the feature dimension $d$ is smaller than the concept retrieval size $K$ (Hastie, 2020). In contrast, lasso can avoid this by sparse regularization. These results indicate that the concept selection by sparse linear regression is crucial in Z-CBMs. In this sense, we can interpret our concept regression as a re-ranking method of the CLIP similarity. Elastic net was the best in accuracy, but it selected more concepts than the other sparse algorithms. This is because elastic net selects all highly correlated concepts to derive a unique solution by combining $\ell_1$ and $\ell_2$ regularization (Hastie et al., 2015). HTP explicitly limits the selected concepts to 256, so while it achieves the highest sparsity, it has the lowest accuracy of the sparse algorithms due to the shortage of concepts for explanation.

### D.5 EFFECTS OF $K$ IN CONCEPT RETRIEVAL

As discussed in Sec. 4, the retrieved concept number $K$ in concept retrieval controls the trade-off between the accuracy and inference time. We assess the effects of $K$ by varying it in $[128, 256, 512, 1024, 2048]$ and measuring the top-1 accuracy and averaged inference time for processing an image. Note that we set 2048 as the maximum value of $K$ because it is the upper bound in the GPU implementation of Faiss (Johnson et al., 2019). Figure 8 illustrates the relationship between the accuracy and total inference time. As expected, the size of $K$ produces a trade-off between accuracy and inference time. Even so, the increase in inference time with increasing $K$ is not explosive and is sufficiently practical since the inferences can be completed in around 55 milliseconds per sample. The detailed breakdowns of total inference time when $K = 2048$ were 0.11 for extracting image features, 5.35 for concept retrieval, and 49.23 for concept regression, indicating that the computation time of concept regression is dominant for the total. In future work, we explore speeding up methods for Z-CBMs to be competitive with the existing CBMs baseline that require training (e.g., Label-free CBMs, which infer a sample in 3.30 milliseconds).

### D.6 EVALUATION ON OUT-OF-DOMAIN DATASETS

To check the generalization capability, we evaluate our Z-CBMs on the out-of-domain test datasets for ImageNet, including ImageNet-V2 (Recht et al., 2019), ImageNet-A (Hendrycks et al., 2021b), ImageNet-R (Hendrycks et al., 2021a), and ImageNet-Sketch (Wang et al., 2019).

Table 11 shows that Z-CBMs outperformed the zero-shot baseline and CDM (except on V2). This suggests that Z-CBMs are robust to domain shifts, while training-based CBMs are susceptible to overfitting.

### D.7 ADDITIONAL QUALITATIVE EVALUATION

We show the additional concept visualization results in Figure 9. We can see the same tendency discussed in Section 5.4. Here, the NOT-prefixed negative concepts represent related but subtly different concepts, like "macro rope," for the objects in the image, which is helpful for users.

Table 11: Top-1 Accuracy (%) on ImageNet OOD datasets.

|  | ImageNet | V2 | A | R | S |
|---|---|---|---|---|---|
| Zero-shot CLIP | 61.88 | 54.60 | 29.89 | 66.79 | 40.81 |
| CDM | 62.52 | **55.93** | 26.32 | 59.15 | 35.83 |
| Z-CBM | **62.70** | 54.82 | **30.27** | **67.56** | **40.94** |

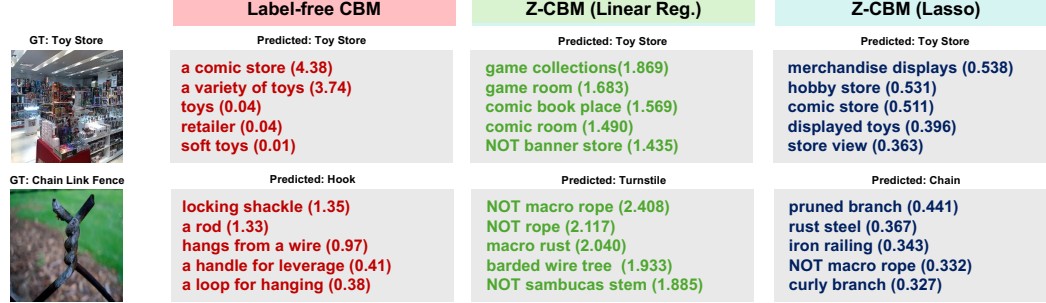

Figure 9: Qualitative evaluation of predicted concepts on the ImageNet validation set. While Label-free CBMs sometimes hallucinate invisible concepts or ignore important concepts, Z-CBMs with lasso consistently provide realistic and dominant concepts in input images with diverse vocabulary. **NOT** prefix denotes that the concept has negative coefficients similar to Oikarinen et al. (2023); Panousis et al. (2023).

### D.8 RELATION BETWEEN CONCEPT BANK AND TARGET DATASET

Here, we investigate how the performance of Z-CBMs depends on the relationship between the concept bank and the target dataset. To evaluate this, we plotted the distribution discrepancy and accuracy difference from black-box baselines for the 12 datasets. As a discrepancy metric, we measured the 2-Wasserstein distance between 100,000 concepts in the bank and class name texts on the CLIP text encoder. Figure 10 shows that the larger the discrepancy, the worse the performance, suggesting that performance may be degraded for domains not covered by the concept bank. In such cases, Z-CBMs can recover performance without training by adding domain-specific vocabulary to the concept bank, e.g., adding GPT-generated concepts for Car recovers the accuracy from 58.87 to 60.12 (Zero-shot CLIP achieves 60.38 as in Table 9).

## E EXTENDED RELATED WORK

**Cross-modal zero-shot classification.** In zero-shot or supervised learning settings, several works (Lampert et al., 2013; Norouzi et al., 2014; Mensink et al., 2014; Jain et al., 2015; Elhoseiny et al., 2013) have explored cross-modal classification methodologies by using textual attributes/concepts as a proxy of image features. ConSe (Norouzi et al., 2014) infers a target label from a semantic embedding composed of a weighted sum of concepts of the single predicted ImageNet label with word2vec embeddings in a fully zero-shot manner. While ConSe is conceptually similar to our Z-CBMs, the zero-shot inference depends on the ImageNet label space, i.e., it cannot accurately predict target labels if there are no target-related labels in ImageNet. In contrast, our Z-CBMs directly decompose an input image feature into concepts via a concept bank, so they are not restricted to any external fixed-label spaces. As a successor work of ConSe, A2C (Demirel et al., 2017) learns input-to-attribute and attribute-to-label mapping by using attributed image datasets for zero-shot inference. While A2C succeeds in outperforming ConSe, the concepts to represent images are restricted to the training datasets, whereas our Z-CBMs are available without additional training and datasets. More recently, Menon & Vondrick (2023) proposed a zero-shot classification method based on the correlation between the input features and the task-specialized texts generated by LLMs for each target class. However, it requires generating the task-specialized texts with LLM and restricting the inference algorithm to the CLIP style zero-shot classification. In contrast, Z-CBMs can be used for arbitrary tasks without external LLMs and arbitrary inference algorithms (e.g., linear probing).

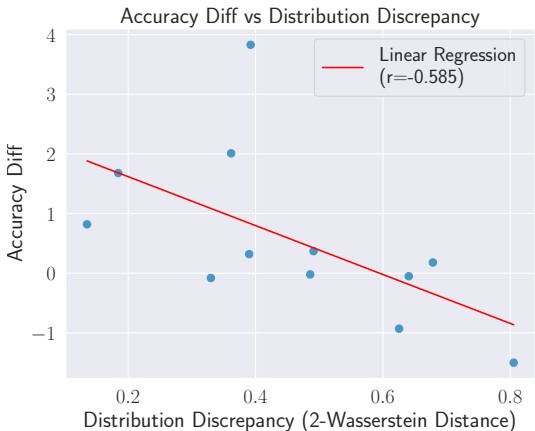

Figure 10: Accuracy difference from zero-shot baseline vs. distribution descrepancy between target class name and concept bank

## F   BROADER IMPACT

This work on Zero-shot Concept Bottleneck Models (Z-CBMs) has the potential for several societal impacts, both positive and negative.

As a positive perspective, by providing explanations for their predictions via human-understandable concepts without requiring task-specific training, Z-CBMs can increase the trustworthiness and transparency of AI systems. This is particularly crucial in high-stakes domains like healthcare (e.g., explaining medical image analysis) or finance (e.g., justifying loan application decisions), where understanding the reasoning behind a decision is paramount. Furthermore, the "zero-shot" nature of Z-CBMs, eliminating the need for extensive training resources and target-specific datasets, can make interpretable AI more accessible to a wider range of researchers, developers, and organizations, including those with limited resources. This could foster innovation and broader adoption of responsible AI practices.

As a negative perspective, the interpretability and intervenability of Z-CBMs could potentially be exploited by malicious users. Understanding which concepts drive a model's decision could make it easier to craft more sophisticated adversarial attacks or to manipulate the model's output by subtly altering input features related to key concepts, perhaps in ways that are hard to detect. Addressing the potential negative impact will require careful consideration during the development, deployment, and regulation of Z-CBMs. This includes rigorously auditing concept banks for bias, developing methods to detect and mitigate manipulation, promoting AI literacy to prevent over-reliance, and establishing clear accountability frameworks.

