# OpenReview forum: "Zero-shot Concept Bottleneck Models"
_ICLR.cc/2026/Conference — ICLR 2026 Conference Withdrawn Submission_

### Official Review · Reviewer_iiKH · 2025-10-30

**Soundness:** 3
**Presentation:** 3
**Contribution:** 2
**Rating:** 4
**Confidence:** 4

**Summary:**

The paper proposes Zero-Shot Concept Bottleneck Models (Z-CBMs), which perform both input-to-concept and concept-to-class inference without any additional training. Z-CBMs leverage a large-scale concept bank, perform concept retrieval (top-K concepts nearest to an image via a frozen vision-language model), and apply sparse linear regression (lasso) to reconstruct the image embedding as a weighted sum of concept embeddings. The resulting concept weights W are then used to predict the final label by comparing the reconstructed feature with label text embeddings. The model requires no finetuning of the underlying VLM.

**Strengths:**

1. A novel zero-shot setting for CBMs, eliminating the need for concept-label annotations and model retraining, which represents a substantial step beyond prior CBM models (e.g., Label-Free CBM, LaBo, CDM).
2. Clear and modular pipeline: Concept retrieval → Concept regression → Label inference.
3. Scalability: The approach leverages CLIP-style embeddings and FAISS indexing for efficient retrieval from million concept vocabulary.
4. Strong empirical coverage: Experiments across 12 datasets show high SigLIP-Scores and strong concept recall compared to training-based CBMs.
5. Intervention demonstration: The authors show that concept deletion/insertion experiments allow human edits of concept sets to meaningfully alter predictions, which supports the intervenability claim.

**Weaknesses:**

1. Limited novelty in method. At its core the method is the combination of concept retrieval + lasso regression, which is straightforward; the primary contribution appears to be the framing of the zero-shot CBM problem rather than a novel modelling technique.
2. Heavy dependence on the VLM (CLIP) feature space. The interpretability and reliability of explanations inherit the biases of the underlying vision-language model; thus, the faithfulness of concept explanations is only as good as CLIP’s embedding geometry.
3. The baseline comparisons raise fairness concerns: The performance of Label-Free CBM and LaBo on CUB/Imagenet reported by the authors appears far lower than in their original papers, making the comparison appear less robust.
4. Although the authors construct a massive concept bank, they do not directly study how to efficiently filter meaningful concept sets; instead they rely on the regression stage to pick important concepts, which may lead to noise being retained.
5. Semantic precision vs. bank size. Collecting 5 million noun phrases may amplify concept noise; there is no thorough ablation or analysis of semantic filtering quality beyond dataset size.
6. Computational cost. Using large-scale FAISS retrieval (K = 2048 per image) and solving lasso regression at inference may be impractical for real-time use or deployment in resource-constrained settings.

**Questions:**

1. How were the top-10 “important” concepts chosen for the Bird (CUB) dataset in Table 1? Table 1’s “concept accuracy” seems much lower than corresponding results in the original CBM paper (Koh et al., 2020). Further, why were stronger CBM variants (e.g., CEMs) not included in the comparison?
2. How close are the concepts retrieved by the model to human-annotated concepts? Is there any quantitative result showing the semantic alignment of retrieved concepts and human label sets?
3. How robust is the concept-retrieval mechanism under domain shift (e.g., medical or satellite imagery)? Would one need domain-specific VLMs or concept banks for such domains?
4. In the regression stage, how should one interpret negative coefficients (e.g., prefixes “NOT” or negative weights)? Do these correspond to counterfactual or suppressive concepts?
5. Table 9 shows that the reported top-1 accuracy improvement of Z-CBM over zero-shot CLIP is very small on average (≈ 0.5 %), and the gains mainly appear on datasets such as Food, SUN, and UCF where class names overlap semantically with the concept bank.
Could the authors analyze whether this improvement primarily comes from lexical overlap between class labels and concept phrases rather than genuine generalization?
A breakdown of performance by the degree of class–concept overlap would clarify this.

---

### Official Review · Reviewer_voqU · 2025-10-31

**Soundness:** 3
**Presentation:** 3
**Contribution:** 2
**Rating:** 4
**Confidence:** 4

**Summary:**

This paper proposes Zero-Shot Concept Bottleneck Models (Z-CBMs) to address zero-shot challenge in CBMs. Traditional CBMs require end-to-end training on target datasets, manual concept annotations (related to class labels), and only allow interventions on static trained concepts, while even VLM-based CBMs still need target-dataset training, restricting practicality. Thus, the paper aims to build models that infer concepts and labels for unseen inputs without additional training. The method of Z-CBMs relies on pre-trained VLMs with frozen weights and a large-scale concept bank. First, the concept bank is constructed by extracting ~5M filtered noun phrases from web caption datasets via NLTK parsing. For concept retrieval (input-to-concept), cross-modal similarity search via Faiss uses the VLM’s image encoder and text encoder to retrieve top-K input-related concepts by cosine similarity. For concept regression (concept-to-class), sparse linear regression approximates the input’s image embedding as a weighted sum of retrieved concept embeddings (with L1 regularization to select essential concepts) and predicts labels by matching the weighted concept embeddings to class-name embeddings, while also supporting flexible interventions on arbitrary natural-language concepts via the VLM’s feature space.

**Strengths:**

- The presentation of this paper is good and the motivation for the method is clearly stated.
- The research topic is very interesting. It breaks out of the normal CBMs pipeline to some extent, adopting a zero-shot setting. I believe this motivation and problem are of great importance.
- The proposed method is simple yet effective, and its presentation is very clear.
- Experiments were conducted on different scale datasets, including general and fine-grained image datasets. The case study in Figure 5 also demonstrates interpretability.

**Weaknesses:**

- In main experiments, under the training setting, the compared methods are somewhat outdated, as they were mostly proposed before 2023. I believe it is necessary to compare with newer SOTA methods. Under the zero-shot setting, the improvement over CLIP zero-shot seems very limited. Only when the concepts from multiple datasets are combined can a more obvious improvement be achieved. And this is still under the condition of using CLIP-ViT-B/32. I think this improvement will further diminish when a large model is used.
- From the Figure 5, in the $W$ of concept regression, the concepts corresponding to elements with relatively high absolute values seem to be well-represented in image $x$. According to CLIP’s zero-shot classification task, $S = {F_{C_x}} ^ T \cdot f_V(x)$ can also represent the presence of concepts in the image, which seems to have the same meaning as $W$. However, numerically, $S \approx {F_{C_x}} ^ T \cdot {F_{C_x}} \cdot W $. Therefore, I am curious about the actual meaning of $W$: does it represent the importance of concepts to the image?
- How many elements in $W$ have negative values? It seems that without regularization, most values are positive, and negative values only appear after regularization is added. Why is this the case? Additionally, many "NOT" concepts in Figure 5 seem to be positively correlated with the image. These negative values left me confused.
- I also have questions about the retrieval of similar concepts in concept retrieval. When there are millions of concepts in your concept bank, it is very normal for similar concepts to be retrieved. This will greatly reduce the number of actually useful concepts in concept retrieval. I want to know, when $K = 2048$, how many of these concepts are actually dissimilar?
- This paper addresses the issue of similar concepts in concept regression. I am curious about the regularization coefficient required to effectively solve the problem of similar concepts.
- I strongly suggest addressing the issue of concept similarity in the first step of concept retrieval. You can refer to the approach in: “ICLR 2024 - Interpreting CLIP's Image Representation via Text-Based Decomposition”.
- In fine-grained image classification, I am not sure whether this method can identify concepts with actual interpretability. Could you provide some examples of top concepts in fine-grained image datasets?
- In $F_{C_x}$, if the dimension $d$ of the representation is smaller than the number of concepts $K$, the rank of this matrix will be less than or equal to $d$. This means there will still be a lot of redundant concepts. Therefore, I do not consider $K = 2048$ a reasonable setup.
- Moreover, if setting $K = d$, making $F_{C_x}$ a square matrix, $W$ can be directly solved as $W = {F_{C_x}} ^ {-1} \cdot f_V(x)$. Would this be more efficient?
- Unrelated to this paper, but just a question: compared with sparse autoencoder (SAE), what do you think are the advantages of the CBM method? The former is unsupervised and does not overly rely on multimodal alignment models.

**Questions:**

See the weakness.

---

### Official Review · Reviewer_bQQa · 2025-11-01

**Soundness:** 2
**Presentation:** 2
**Contribution:** 2
**Rating:** 4
**Confidence:** 4

**Summary:**

The paper proposes Zero-shot Concept Bottleneck Models (Z-CBMs): retrieve concept candidates from a very large concept bank with CLIP, then select sparse, non-redundant concepts via lasso to reconstruct the image embedding and predict labels. Experiments on 12 datasets show competitive accuracy and interpretable, intervenable concepts without task-specific training.

**Strengths:**

1. Retrieval + sparse linear regression is well-motivated and easy to implement on top of existing VLMs.

2. Mining ~5M noun phrases from web captions broadens coverage beyond class-conditioned concept lists; FAISS makes retrieval practical.

3. Sparse regression reduces semantic duplication; negative coefficients are exposed as “NOT” concepts; concept deletion/insertion studies are a good step.

**Weaknesses:**

1. CLIP-Score (computed in the same embedding family) is used to evaluate concept quality, risking self-confirmation. Human judgments or model-agnostic faithfulness tests (e.g., concept occlusion, counterfactual patching, completeness/sufficiency) are missing.
2. Approximating an image embedding as a weighted sum of text embeddings is intuitive but under-theorized. There’s no analysis of whether CLIP’s joint space is approximately text-spanned, how normalization affects regression, or when negative coefficients are semantically meaningful.
3. If class names or near-synonyms exist in the concept bank, retrieval/regression may implicitly re-use labels. The paper does not systematically exclude label terms or quantify the effect of removing them (and close paraphrases).
4. Intervention is demonstrated mainly on CUB with attribute labels, and insertion uses linear regression instead of lasso (changing the mechanism mid-experiment). No user study or broader dataset evidence is provided.

5. Missing latency/memory figures for building and querying a 5M-vector FAISS index and solving lasso over K=2048 concepts per image. Practical on-device feasibility and throughput are unclear.

6. Noun-phrase extraction may introduce ambiguous or low-quality concepts; filtering is deferred to an appendix without descriptive statistics (e.g., proportion removed, harmful/unsafe phrases). Diversity/coverage metrics beyond overlap with another CBM are limited.

7. It’s unclear whether text vectors are standardized before lasso (important for feature-scale invariance) and whether the similarity of reconstructed embeddings is re-normalized prior to label scoring. Hyperparameter selection (e.g., λ=1e-5) seems lightly justified.

**Questions:**

See weakness part

---

### Official Review · Reviewer_Lf8Y · 2025-11-05

**Soundness:** 3
**Presentation:** 2
**Contribution:** 2
**Rating:** 4
**Confidence:** 3

**Summary:**

CBMs traditionally require human annotations in form of concepts to train concept explanations for a label. This paper introduces Zero-shot CBMs where one doesnt require expensive concept annotations. ZCBM uses VLM and a large concept bank (5 million) extracted from web-scale image caption datasets. The framework has 2 stages: concept retrieval: which dynamically identifies relevant concepts via cross-modal similarity search and concept regression: sparse linear regression (lasso) to select essential concepts by reconstructing input image features as weighted combinations of concept embeddings. They perform experiments across 12 datasets and show strong image concept correlation.

**Strengths:**

- The problem is well motivated and eliminates the key bottlenecks of CBMs: concept annotation and training.

- The authors performed extensive experiments across 12 diverse datasets with ablation experiments.

**Weaknesses:**

- For the datasets, human concept annotations exists, there is no comparison against standard/best performing CBM/CEM. I think it is an important point of discussion whether LLM concept annotations are better than human concept annotations.

- 5M concept embeddings: there is not much detail on what these concepts are. I think it is safe to assume that this concept dictionary is bringing in boost in the performance. I would have appreciated more details on this.

**Questions:**

- Fig 3 and 4 are cut, please update them

- Is it possible to bring the Table 8 into the main paper to understand the context of the results better?

- how does final performance change with retrieval recall @k?

- Why does Fig 3 have Concept Deletion Ratio and Fig 4 Number of Inserted Concepts/Sample instead of % Concept insertion? CBMs usually plot until all of the concepts are identified.

---

### Note · Authors · 2025-11-12

**Comment:**

Dear Reviewers,

We sincerely thank you for taking the time to provide us with so many valuable comments. We are delighted that the reviewers recognized the fundamental motivation and value of this paper. On the other hand, the fact that we did not receive recommendations for acceptance from any reviewers made us realize that this paper has room for improvement. Therefore, we have decided to withdraw this paper and strengthen it based on your comments before resubmitting it to another conference. Finally, we would like to express our sincere gratitude once again for your detailed and insightful comments.

Best regards,

Authors

**Withdrawal Confirmation:**

I have read and agree with the venue's withdrawal policy on behalf of myself and my co-authors.